# Retinoid X Receptor Activation Prevents Diabetic Retinopathy in Murine Models

**DOI:** 10.3390/cells12192361

**Published:** 2023-09-26

**Authors:** Iuliia Dorofeeva, Assylbek Zhylkibayev, Irina V. Saltykova, Venkatram Atigadda, Bibek Adhikari, Oleg S. Gorbatyuk, Maria B. Grant, Marina S. Gorbatyuk

**Affiliations:** 1Department of Optometry and Vision Science, School of Optometry, University of Alabama at Birmingham, Birmingham, AL 35233, USA; dorofeeva@uab.edu (I.D.); askokshe@uab.edu (A.Z.); irina.saltykova@bcm.edu (I.V.S.); adhikari@uab.edu (B.A.); oleggor@uab.edu (O.S.G.); 2Department of Dermatology, Heersink School of Medicine, University of Alabama at Birmingham, Birmingham, AL 35233, USA; venkatra@uab.edu; 3Department of Ophthalmology and Vision Sciences, Heersink School of Medicined, University of Alabama at Birmingham, Birmingham, AL 35233, USA; mariagrant@uabmc.edu

**Keywords:** retinoid X receptor, diabetic retinopathy, murine model, UAB126

## Abstract

Previously, the RXR agonist UAB126 demonstrated therapeutic potential to treat obese mice by controlling blood glucose levels (BGL) and altering the expression of genes associated with lipid metabolism and inflammatory response. The purpose of the study was to assess the effects of UAB126 on the progression of diabetic retinopathy (DR) in rodent models of type 1 diabetes (T1D), streptozotocin-induced, and type 2 diabetes (T2D), in db/db mice. UAB126 treatment was delivered either by oral gavage for 6 weeks or by topical application of eye drops for 2 weeks. At the end of the treatment, the retinal function of diabetic mice was assessed by electroretinography (ERG), and their retinal tissue was harvested for protein and gene expression analyses. Bone-marrow cells were isolated and differentiated into bone marrow-derived macrophages (BMDMs). The glycolysis stress test and the 2-DG glucose uptake analysis were performed. Our results demonstrated that in the UAB126-treated diabetic BMDMs, the ECAR rate and the 2-DG uptake were improved as compared to untreated diabetic BMDMs. In UAB126-treated diabetic mice, hyperglycemia was reduced and associated with the preservation of ERG amplitudes and enhanced AMPK activity. Retinas from diabetic mice treated with topical UAB126 demonstrated an increase in Rxr and Ppar and the expression of genes associated with lipid metabolism. Altogether, our data indicate that RXR activation is beneficial to preclinical models of DR.

## 1. Introduction

Functioning as a nuclear receptor (NR), the retinoid X receptor (RXR) is a transcriptional factor that binds to the promoter region of genes either as a dimer with itself (homodimer) or with another NR to form a heterodimer. Although RXR can activate transcription as a homodimer, this receptor also serves as an obligatory common dimerization partner for numerous other NRs [1,2]. The RXR-NR heterodimer binding sequence consists of two six-base pair sequences [3].

In general, the molecular structure of RXR is like that of other NRs; RXR proteins contain an A/B region located at the N-terminal, a DNA-binding domain (DBD), and a ligand-binding domain (LBD) positioned at the C-terminal. The role of the A/B domain, while still under investigation, is believed to attract specific binding partners, which determine either the activation or repression of the transcriptional complexes [4]. Moreover, nuclear localization sequences are present within the DBD domain of RXR proteins [5], and in the absence of ligands, RXR can be found both in the cytoplasm and in the nucleus [6].

RXR occupies a critical position in the NR family because of its ability to form heterodimers with many other family members and is therefore involved in controlling a myriad of physiologic processes. RXR heterodimers can be classified into two main groups: permissive and non-permissive binding partners. Permissive heterodimers are those that can be activated by the binding of RXR agonists, binding of the agonists of the NR partner, or binding of the agonist to both NRs [7]. Examples of this type of heterodimerization include the RXR/liver X receptor (LXR), the RXR/peroxisome proliferator-activated receptor (PPAR), and the RXR/farnesoid X receptor (FXR) heterodimers [8]. The binding of ligands to both partners could provide an additive or synergistic biological response. Non-permissive heterodimers require that the agonist binds to the heterodimerization partner, while RXR acts as a silent partner. Examples of this heterodimerization include RXR/RAR (retinoid A receptor), RXR/VDR (vitamin D receptor), and RXR/TR (thyroid hormone receptor).

Overall, three different isoforms of RXR—RXRα, RXRβ, and RXRγ—exist. *RXRα* and *RXRβ* mRNAs are widely expressed, whereas *RXRγ* mRNA is restricted to only a few tissues, including the retina [9,10,11]. RXRα is a predominant isoform that has been identified in the nucleus, cytoplasm, and mitochondria [12]. RXRβ is abundantly expressed in endothelial cells and cancer cell lines and is largely conserved across species [13]. RXRγ is a 14 kB-long isoform containing nine introns widely ranging in size. High RXRγ expression is found in the brain, and its expression is important for the development of memory deficits and depression-like behavior upon its ablation [14].

Mounting evidence indicates that abnormal RXR signaling is involved in neuronal stress and neuroinflammatory responses in several neuropathological conditions. RXR protective effects have been established in various cell and animal models, including Alzheimer’s, Parkinson’s, glaucoma, multiple sclerosis, stroke [13], colitis [15], pulmonary emphysema [16], and rheumatoid arthritis [17], by modulating inflammatory responses [18]. In addition, RXRs also control the clearance of apoptotic cells and β-amyloid protein by macrophages; the lack of RXRα impairs the transcription of genes associated with phagocytosis-related events [19].

Another physiological role of RXR is the regulation of lipid metabolism. Bexarotene, an FDA-approved rexinoid, improved cholesterol homeostasis and inhibited atherosclerosis progression in a mouse model of dyslipidemia [20]. The level of hepatic gene expression for *Scd1* (stearoyl-Co enzyme A desaturase 1), *Fas* (fatty acid synthase), *Angptl3* (angiopoietin-like 3), *Abaca1* (ATP-binding cassette subfamily a member 1), and *ApoA-1* (apolipoprotein A1) was significantly increased in these mice, suggesting improved cholesterol efflux. Despite this fact, RXR activation can increase the expression of SREBP-1, leading to elevated triglycerides [20].

Recently, a novel rexinoid-like molecule, UAB126, has been developed [21,22]. The oral administration of UAB126 reduced obesity, insulin resistance, and dyslipidemia without changing thyroid hormones [22]. Treatment of mice with UAB126 significantly increased expression of the genes responsible for lipid metabolism and decreased inflammatory gene expression in white adipose tissue. Interestingly, UAB126 treatment did not raise serum triglyceride levels and also improved insulin signaling via a decrease in overnight blood glucose and insulin levels [22].

Overall, the role of RXR in the diseased retina has not been explored carefully, with the exception of the study with rd1 mice that manifested inherited retinal degeneration, or retinitis pigmentosa [10], which showed that the RXR agonist PA024 decreased photoreceptor cell death [10].

Given that RXR could be a therapeutic target for degenerating retinas, that UAB126 controls lipid metabolism and inflammatory response in obese mice, and that dyslipidemia and inflammation are two critical components of diabetic retinal pathogenesis, we tested UAB126 in two models of DR and demonstrated the therapeutic potential of UAB126 to treat diabetic retinal dysfunction and delay the onset of DR in mice.

## 2. Materials and Methods

### 2.1. Animals

Male C57BL/6J (Strain#: 000664) mice and db/db (BKS.Cg-Dock7m +/+ Leprdb/J; Strain#: 000642) mice were obtained from The Jackson Laboratory (Bar Harbor, ME, USA) and housed in the University of Alabama at Birmingham (UAB) animal facility, adhering to the guidelines set by the Institutional Animal Care and Use Committee and the Association for Research in Vision and Ophthalmology. The mice were kept in a 12 h light/dark cycle with ad libitum access to food and water.

### 2.2. Diabetes Induction

At 8 weeks of age, diabetes was induced by administering five consecutive intraperitoneal injections of 50 mg/kg streptozotocin (STZ) or vehicle (10 mM sodium citrate buffer, ice-cold, pH 4.5). Prior to each STZ injection, the mice underwent a 6 h food deprivation. Animals were considered diabetic when their blood glucose concentrations exceeded 250 mg/dL on two separate measurements a minimum of two days apart. Throughout the study period, blood glucose levels were monitored weekly, and HbA1C and insulin levels were measured at the time of euthanasia. The oral glucose tolerance test was performed as previously described [23]. Animals were euthanized by carbon dioxide following cervical dislocation.

### 2.3. UAB126 Synthesis

The synthesis of UAB126 has been performed as previously described [22].

### 2.4. Preparation of UAB126 Eye Drops

The nano emulsion development was carried out using the phase inversion temperature method, as described in the references by Singh et al. [24] and Fernandes et al. [25]. In this process, a surfactant mixture comprising polysorbate-80 (200 mg) and pluronic-127 (62.5 mg) was heated to a temperature above 70 °C and mixed using a magnetic stirrer at 250 rpm for 10 min. Once a clear jelly-like liquid was obtained, the active pharmaceutical ingredient UAB-126 (150 mg) was added, and mixing continued until a clear and transparent jelly was formed. Simultaneously, in a separate beaker, 25 mL of water was heated to 70 °C, and beta-cyclodextrin (125 mg) and EDTA (6.25 mg) were dissolved in it. This solution was then used to solubilize the UAB-126 surfactant mixture. Finally, the pH of the solution was adjusted to 7.4 using either 4M sodium hydroxide or diluted hydrochloric acid, as required.

### 2.5. UAB126 Systemic Administration

UAB126 was administered orally in 10% DMSO mixed with corn oil at a dosage of 100 mg/kg, starting a month after the STZ injection, and continued daily for 6 weeks.

### 2.6. UAB126 Topical Application

Ten-week-old db/db animals were used in this experiment. One eye was treated with 10 µL of UAB126 eye drops, and the other eye received the same amount of PBS as a Control. Eye drops were applied every day for 2 weeks. After 2 weeks, retinas were collected for LC-MS/MS study to detect UAB 126 after the treatment and for RNA isolation following qRT-PCR analysis and immunohistochemistry.

### 2.7. Retinal Explants

The eyes from C57BL6J animals at postnatal day 8 were enucleated, and the retina was gently isolated from the eyecup and placed in neurobasal serum-free medium (Neurobasal-A, 10888022; Invitrogen, Carlsbad, CA, USA) containing 2% B27 (0080085-SA; Invitrogen, Carlsbad, CA, USA), 1%N2 (17502-048; Invitrogen), 2 mM GlutaMAX (35050038; Invitrogen), and 100 units/mL penicillin–100 lg/mL streptomycin (P4333; Sigma-Aldrich Corp., St. Louis, MO, USA). Retinal explants were maintained at 37 °C in a 5% CO_2_ condition. Mannitol (19 mM; control) (Sigma, M4125) and D-glucose (34 mM; high glucose) (Sigma, G5767) were dissolved in growth medium. Rexinoid 3, Rexinoid 6, and UAB126 were prepared in sterile water and added to treatment groups with a final concentration of 100 µM. Explants were cultured for 24 h.

### 2.8. Quantitative Real-Time PCR Analysis (qRT-PCR)

Total RNA from retinas was isolated using TRIzol reagent (15596018, Invitrogen, Grand Island, NY, USA). RNA (1 μg) was then reversely transcribed into cDNA using SuperScript IV Master Mix with ezDNase (11766050, Thermo Fisher Scientific, Waltham, MA, USA). A qRT-PCR was performed using the Quant Studio 3 (Applied Biosystems, Foster City, CA, USA) system with TaqMan primers (Thermo Fisher Scientific, Waltham, MA, USA). Following is a list of genes with reference numbers from Thermo Fisher Scientific Company: Elovl4-Mm00521704_m1; Acly-Mm01302282_m1; Fasn-Mm00662319_m1; Scd1-Mm00772290_m1; Acc1-Mm01304258_m1; GAPDH-Mm99999915_g1. Lxr beta was purchased from Biorad (Nr1h2-Cat#10031252). The following cycling protocol was used: 95 °C for 20 s, followed by 40 cycles of amplification at 95 °C for 15 s and 60 °C for 60 s.

Sybr green real-time PCR reaction was performed using the next primers: Rxr alpha (forward5′CATTGGGCTTCGGGACTGGT3′, reverse 5′CCTCGTTCTCATTCCGGTCC3′); Rxr beta (fwd5′ATTCCTCCGGGCCTGTCAGCA3′, rev5′CTCCATCCCCGTCTTTGTCC3′); Rxr gamma (fwd5′CAGGTCTGCCTGGGATTGGA3′ REV5′CCTCACTCTCTGCTCGCTCT3′); PPAR alpha (fwd5′AGAAGTTGCAGGAGGGGATT3′, rev5′TTGAAGGAGCTTTGGGAAGA3′); PPAR gamma (fwd5′CCCCTACAGAGTATTACG3′, rev5′TCTCTCCGTAATGGAAGACC3′); GAPDH (fwd5′TGACGTGCCGCCTGGAGAAA3′, rev5′AGTGTAGCCCAAGATGCCCTTCAG3′). PCR cycling program: 95 °C for 2 min 20 s, followed by 40 cycles of amplification at 95 °C for 20 s and 60 °C for 60 s. Relative expression levels of genes were calculated using the ΔΔCt method.

### 2.9. Tissue Homogenization and Extraction for Mass Spectrometry

Kidney, retina, and brain tissues were weighed and homogenized at a density of 140 mg of dry tissue per milliliter in ethyl acetate/isopropanol (4:1) using a Bead Mill 4 homogenizer (Thermo Fisher Scientific, Waltham, MA, USA) and 2 mL pre-filled polypropylene microtubes (2.8 mm ceramic beads, 3 × 20 s cycles, speed 5). Insoluble debris was removed by centrifugation (10,000× *g*, 30 min, 5 °C). Fixed aliquots of the supernatants (225 μL) were transferred to clean Eppendorf tubes and evaporated under a gentle stream of nitrogen gas. The residues were reconstituted in 75 μL of methanol/water (4:1), centrifuged (18,000× *g*, 30 min, 5 °C), and transferred to 2 mL autosampler vials equipped with low-volume polypropylene inserts and Teflon-lined rubber septa for analysis.

### 2.10. Mass Spectrometry

Retinal samples after systemic UAB126 delivery (oral gavage) and topical application (eyedrops) were sent for LC-MS/MS study. The LC-MS/MS analysis was performed using a Thermo Quantum Ultra triple quadrupole mass spectrometer interfaced with a Waters Acquity UPLC system (Waters Corp., Milford, MA, USA). The mass spectrometer was operated in negative ion mode with the following optimized APCI source parameters: corona discharge current 22 μA; ion transfer tube temperature 300 °C; vaporizer temperature 290 °C; N_2_ sheath gas 40; N_2_ auxiliary gas 5; in-source CID 14. Quantitation was based on MRM detection (UAB-126: *m*/*z* 273 → 229, collision energy 15, tube lens 90; Ar collision gas 1.5 mTorr; scan time 100 ms; Q3 scan width 1.0 *m*/*z*; Q1/Q3 peak widths at half-maximum 0.7 *m*/*z*.). Data acquisition and quantitative spectral analysis were done using Thermo Xcalibur version 2.0.7 SP1 and Thermo LCQuan version 2.7, respectively. Calibration curves were constructed by plotting peak areas against analyte concentrations for a series of nine calibration standards, ranging from 0.375 to 3750 nmol total analyte. A weighting factor of 1/C2 was applied in the linear least-squares regression analysis to maintain homogeneity of variance across the concentration range. A kinetic XB-C18 reverse phase analytical column (2.1 × 100 mm, 2.6 μm, Phenomenex, Torrance, CA, USA) was used for all chromatographic separations. Mobile phases were made up of 15 mM ammonium acetate and 0.2% acetic acid in (A) water/acetonitrile (9:1) and in (B) acetonitrile/methanol/water (90:5:5). Gradient conditions were as follows: 0–1.0 min, B = 40%; 1–5 min, B = 40–100%; 5–5.5 min, B = 100%; 5.5–6 min, B = 100–40%; 6–10 min, B = 40%. The flow rate was maintained at 300 μL/min, and the total chromatographic run time was 10 min. A software-controlled divert valve was used to transfer the LC eluent from 0 to 1.5 min of each chromatographic cycle to waste.

### 2.11. Mouse Bone Marrow-Derived Macrophages (BMDM) Generation

Bone marrow cells from the healthy control and diabetic UAB126-treated and untreated mice were isolated by centrifuging bones at >10,000× *g* in a microcentrifuge tube for 15 s, as previously described [26]. Briefly, we isolated bone-marrow-derived cells, grew them in complete DMEM medium with 10% FBS and 1 IU/mL Pen-Strep, and then differentiated the cells into macrophages by supplementing with M-CSF (20 ng/mL, Peprotech, Rocky Hill, NJ, USA). Seahorse stress testing was performed to compare their metabolic glycolytic activity after 7 days of differentiation.

### 2.12. BMDM Activation and Pro-Inflammatory Assessment

On the 7th day of culturing, BMDM were detached using 15 mM EDTA in PBS for 10 min on ice. The macrophage population was then evaluated by flow cytometry, specifically by examining the expression of CD45/CD11b (double positive) and F4/80 markers, which yielded a percentage of >96%. Subsequently, the detached cells were seeded in 12-well plates and cultured in DMEM supplemented with 10% FBS and 1 IU/mL pen-strep. The following day, non-adherent cells were washed away with PBS, leaving the macrophages behind. The activation process involved adding 200 ng/μL of LPS to the culture media. After 24 h of conditioning, the media was collected for measurements of pro-inflammatory cytokines using the V-PLEX Proinflammatory Panel 1 Mouse Kit by Meso Scale Diagnostics (K15048D-1, Rockville, MD, USA).

### 2.13. Glucose Uptake Assay

One day before the assay, BMDM cells were seeded in a 24-well plate at a density of 3 × 10^4^ cells/well. The assay was conducted using flow cytometry. Positive control cells were treated with phloretin for 1 h before the assay. The assay involved exposing the cells to fluorescent 2-deoxy-2-[(7-nitro-2,1,3-benzoxadiazol-4-yl) amino]-D-glucose (2-NBDG) for 30 min, following the manufacturer’s recommendations (ab287845, Abcam, Waltham, MA, USA). For the flow cytometry analysis, cells were washed, trypsinized, and resuspended in 400 µL of assay buffer. Glucose uptake was measured using an LSR Fortezza flow cytometer at the UAB Comprehensive Flow Cytometry Core.

### 2.14. Metabolic Stress Test (Seahorse Assay)

Resuspended cells were plated in the XF96 tissue culture microplate (Agilent Technologies, Seahorse Bioscience, Santa Clara, CA, USA) at a concentration of 3 × 10^5^ cells per well in 180 μL of growth media (DMEM with 5.56 mM glucose, 1.0 mM pyruvate, 4 mM L-glutamine, 10% FBS, and 1% pen-strep). On the day of the assay, the growth media was removed, and the cells were washed once and replaced with either glycolytic stress test media (phenol-red-free DMEM, with 4 mM glutamine, pH 7.4 at 37 °C) or mito stress test media (phenol-red-free DMEM, with 5.56 mM glucose, 1 mM pyruvate, 4 mM glutamine, pH 7.4 at 37 °C) to achieve a final volume of 180 μL per well for their respective tests. The cells were then incubated at 37 °C in a non-CO_2_ incubator for 1 h and subsequently loaded into the Seahorse XFe96 Analyzer (Agilent Technologies, Seahorse Bioscience) when prompted. The Seahorse XF96 sensor cartridge (Agilent Technologies, Seahorse Bioscience) was hydrated in sterile water in a non-CO_2_ 37 °C incubator overnight and then in pre-warmed calibrant in a non-CO_2_ 37 °C incubator for 1 h before the assay run. All effectors were prepared to 10× desired concentrations and loaded into the appropriate ports of the sensor cartridge at 20, 22, and 25 µL in ports A, B, and C, respectively, for the glycolytic stress test assay (port A: 100 mM glucose, port B: 15 mM oligomycin, port C: 500 mM 2-DG), and the mito stress test assay (port A: 15 µM oligomycin, port B: 10 μM FCCP, port C: 5 µM each rotenone and antimycin A). The cartridge was then loaded into the Seahorse XFe96 Analyzer, and the protocol was run as follows: calibrate (15 min), equilibrate (15 min), 3× basal reads (mix: 3 min, wait: 0 min, measure: 3 min), followed by a port injection (A, B, C), each followed by 3× reads. The extracellular acidification rate (ECAR) was obtained at baseline and in response to each effector injection.

### 2.15. Electroretinography (ERG)

The protocol consisted of consecutive scotopic and photopic ERGs (UTAS BigShot ERG instrument, LKC Technologies, Gaithersburg, MD, USA). For scotopic ERG, the mice were dark-adapted for 12 h, and all subsequent procedures were conducted in a dark room. Anesthesia was induced with an intraperitoneal injection of ketamine/xylazine based on the mice’s weight. Topical 2.5% phenylephrine (Paragon BioTeck, Inc., 42702–102-15, Portland, OR, USA) and Gonak (2.5% sterile hypromellose ophthalmic demulcent solution, AKORN, Lake Forest, IL, USA) were applied to the corneal surfaces. Next, a monopolar contact loop was positioned on the cornea to record the retinal a- and b-wave ERG amplitudes. For scotopic ERG, the mice were exposed to 25 flashes of white LED light with an intensity of 3.14 × 10^−5^ cds/m^2^, with 1.5 s intervals between flashes. Subsequently, a series of 5 flashes was applied at various intensities: 0.025 cds/m^2^, 2.5 cds/m^2^, 7.91 cds/m^2^, and 25 cds/m^2^, with 45-s intervals between flashes. Following the scotopic protocol, each mouse was light adapted for 10 min under a dome background light of 25 cd/mm. The photopic protocol involved a series of 15 flashes with 1-s intervals between each flash, at intensities of 2.5 cds/m^2^, 7.91 cds/m^2^, 25 cds/m^2^, and 79 cds/m^2^. The resulting a- and b-waveforms were analyzed using the LKC EM (LKC Technologies, Gaithersburg, MD, USA).

### 2.16. Immunoblotting

For immunoblotting, mouse retinas were isolated and homogenized using RIPA buffer supplemented with 1% Halt Protease Inhibitor and a phosphatase inhibitor cocktail (Thermo Fisher Scientific, Waltham, MA, USA) and prepared for Western blot as previously described [27]. Protein samples (40–60 μg) were separated by SDS-PAGE and transferred to a PVDF membrane. The anti-phosphorylated AMPK (Cell Signaling #2535T, dilution 1:1000, Beverly, MA, USA) and anti-AMPK (Cell Signaling, #5832S, dilution 1:1000 Beverly, MA, USA) antibodies were used for target protein detection.

### 2.17. Immunohistochemistry

To perform immunohistochemistry (IHC), the eyes were fixed with 4% paraformaldehyde for 4 h and subsequently preserved in a series of sucrose solutions (15% and 30% sucrose). After preservation, the eyes were washed in 1x PBS and embedded in Tissue-Tek O.C.T. compound (#4583, Sakura Fintek USA, Torrance, CA, USA). The embedded eyes were then frozen in 2-methylbutane at −43 °C. IHC analysis was carried out on 12 μm-thick retinal sections. The primary anti-RXR antibody (Invitrogen, #433900, dilution 1:200 Waltham, MA, USA) was used to detect RXR in the human and mouse retinas. Fluorescent confocal microscopy with a Nikon AX-R was performed to take images of retinal sections.

## 3. Results

### 3.1. RXR Is Reduced in the Diabetic Retinas of Humans and Mice

Immunohistochemical analysis of both human and mouse T2D diabetic retinas (Figure 1) depicts reduced RXR staining as compared to age-matched controls (human non-DM and db/m mice). Retinal pigment epithelial (RPE), ganglion (RGC), and photoreceptor (Rh) cells showed a marked reduction in RXR.

### 3.2. UAB126 Is Present in the Serum and Retina Following Oral Administration

We next analyzed UAB126 levels by the LC-MS/MS method in the serum and retinas of mice given UAB126 by gavage. Samples were collected 2 h later (Figure 2A). Both the serum and retinal samples had peaks corresponding to the UAB126-loaded standard, while the samples from the vehicle-treated mice showed no comparable peaks. These data demonstrate that UAB126 can stay non-metabolized in diabetic mice for at least 2 h and reach the retina upon systemic treatment.

### 3.3. UAB 126 Treatment Reduces Hyperglycemia in Diabetic Mice

Treatment of diabetic mice with UAB126 significantly reduced blood glucose levels, although not to normal (Figure 2B). In addition, we found that glucose tolerance was reduced in UAB126-treated diabetic mice at 2 h as compared to the untreated group (Figure 2C). In contrast to the improved glucose levels (Figure 2D), insulin did not change in UAB 126 diabetic mice (Figure 2E). This suggests that the treatment primarily regulates insulin receptor levels, resulting in enhanced glucose uptake. Both UAB126-treated and untreated diabetic mice demonstrated no differences in body weight (Figure 2F).

### 3.4. UAB126 Treatment Improves the Diminished Retinal Function and Alters Cellular Signaling in Diabetic Retina

Mice were analyzed by scotopic ERG analysis at 6 weeks of treatment. Figure 3A,B demonstrate that while vehicle-treated diabetic mice showed a reduction in ERG amplitudes, treatment with UAB 126 markedly prevented the decline of A- and B-waves. It is also worth noting that although the B-waves were significantly higher in UAB126-treated vs. untreated groups, the amplitudes did not reach the level of those found in the healthy control group, suggesting that UAB126 treatment delays the progression of vision loss in diabetes. Western blots of retinas from the experimental cohorts showed a significantly higher level of p-AMPK as compared to vehicle-treated mice.

### 3.5. UAB126 Regulates Metabolism-Relative Biological Pathways in Diabetic Retinas

During DR progression, the diabetic retina experiences monocyte infiltration, which is critical for the overall innate immune response and inflammation. Therefore, we next assessed the properties of BMDM macrophages from UAB126-treated mice. BMDM then underwent a glycolysis stress test by measuring the extracellular acidification rate (ECAR) using the Seahorse approach. Figure 4A demonstrates that the diabetic macrophages derived from vehicle-treated mice showed an increase in the acidification rate after the addition of glucose, oligomycin, and 2-DG as compared to the healthy control group; UAB126 treatment dramatically reduced the ECAR in diabetic macrophages, making their ECAR indistinguishable from healthy macrophages. These data demonstrate that glycolysis in vehicle-treated diabetic macrophages may be enhanced, leading to an increase in lactate efflux. Therefore, we next asked ourselves whether the increase in ECAR could be associated with an increase in glucose uptake by macrophages.

Indeed, a significant (over 2-fold) increase in the uptake of 2-DG labeled with Alexa Flour 488 was detected in diabetic BMDM from vehicle-treated mice (Figure 4B). Interestingly, treatment with UAB126 markedly reduced the 2-DG uptake in macrophages. We then analyzed the pro- and anti-inflammatory cytokines in treated macrophages and found that UAB126 treatments led to an increase in the IL-10 cytokine, while Il-1β was not different between groups of macrophages (Figure 3C). Overall, the results of the study indicate that UAB216 treatment influenced glucose metabolism in diabetic macrophages as compared to untreated diabetic cells, suggesting that systemic delivery of UAB126 not only reduced BGL and normalized glucose metabolism, leading to improved retinal electrophysiology, but also increased the anti-inflammatory program in diabetic macrophages.

### 3.6. Topical Delivery of UAB 126 Activated RXR in Diabetic Retinas

It is worth mentioning that drugs with well-recognized hypoglycemic action, when administered topically in db/db mice, do not reduce blood glucose levels, and their effects are assessed directly [28,29]. Therefore, evaluating the systemic effect of UAB126 treatment on diabetic mice, we wondered whether topically applying eye drops containing the rexinoid could be an effective drug delivery approach. Ocular application of UAB126 drops on the corneal surface of db/db mice was carried out daily for two weeks.

The LC-MS/MS results demonstrate that the control (untreated) retinas did not contain a UAB126-associated peak and that a strong signal was detected in the retina treated with UAB126 drops (Figure 5A). Moreover, RXR staining of the treated retina with anti-RXR antibody demonstrated strong immunoreactivity as compared to buffer-treated retinas (Figure 5B). The increase in the *RXRα* mRNA expression level was also detected in the treated eyes, which was much higher than in untreated diabetic and healthy control group eyes (C57BL6) (Figure 5C). Interestingly, like RXR, PPAR was also reduced in diabetic retinas, and UAB126 treatment also increased its expression level. In addition to the altered expression of two NRs, RXR and PPAR, the expression of genes responsible for lipid metabolism, such as *Elovl4*, *Acly*, *Fasn*, *Scd1*, and *Acc1*, was reduced in untreated diabetic retinas and increased upon UAB126 treatment *(p* < 0.05), with the exception of the *Scd1* and *Acc1* genes.

We next decided to assess the ability of other rexinoids to increase PPAR and LXR expression in an ex vivo experiment. Figure 5D shows that the retinal explants cultured in the media with high glucose (to model diabetic retinopathy) and supplemented with rexinoid 3 and rexinoid 6 [30] demonstrated significant upregulations of *Rxrα*, *Rxrβ*, and *Rxrγ* mRNAs at 24 h. We next examined the expression of the *Pparα* and *Pparγ* genes and found significant increases in their expression as well.

## 4. Discussion

The main findings of this study include that diabetic human and mouse retinas exhibit reduced levels of RXR expression in the ONL, INL, RGL, and RPE. UAB126 preserves the A-wave amplitude and partially protects the B-wave amplitude. This indicates the protection of retinal function, an effect that could be mediated by the reduction in glucose in the UAB126-treated diabetic mice. In vitro studies with bone marrow-derived macrophages from diabetic untreated and UAB-treated mice demonstrate that UAB126 treatment increased expression of the anti-inflammatory cytokine IL-10. Direct ocular delivery in T2D mice using topical UAB126 application resulted in an increase in retinal mRNA for Rxrα, Pparα, and lipid-associated gene expression. Collectively, our data support the beneficial effect of RXR agonism on the neural retina by modulating glucose metabolism and increasing expression of the key anti-inflammatory cytokine IL-10. These findings suggest that RXR represents a novel target for DR treatment.

The previous study in obese mice showed that UAB126 can control glucose levels; both glucose and HbA1C levels were reduced in our study. These data imply that UAB126 could be beneficial to diabetic patients not only by maintaining blood glucose and therefore managing the progression of diabetes but also by directly altering the biological processes in diabetic retinas. Our studies suggest that UAB126 treatment reprograms cellular metabolism in macrophages, modifying glycolysis and glucose consumption rates. The UAB126-treated macrophages responded to all steps of metabolic stress, similar to cells isolated from healthy mice. The response of macrophages to metabolic stress has been controversial. Zeng et al. proposed that glycolysis is upregulated in C57BL6 BMDM under high glucose conditions [31] and the ECAR rate is enhanced in adipose tissue macrophages derived from obese mice [32,33]. In contrast, peritoneal macrophages isolated from STZ-induced diabetic Ldr^−/−^ mice manifest suppressed glucose uptake and glycolysis [34], and the results by Pavlou et al. proposed that long-term exposure of C57BL6 BMDM in medium supplemented with high glucose suppresses the ECAR rate [35]. The controversy could be explained by differences in the mouse strains, sources of isolated macrophages, the conditions of treatment, and in vitro vs. in vivo glucose exposures. However, the importance of metabolic shifts in diabetic macrophages is widely recognized. It has been proposed that M1 macrophages rely mainly on glycolysis and that their aberrant TCA cycle leads to the accumulation of succinate, which in turn induces HIF1α stabilization and activates the transcription of glycolytic genes, which sustain glycolytic metabolism [36]. In contrast, M2 macrophages depend on oxidative phosphorylation, and their TCA cycle is intact. Together, these data suggest that the metabolic state of macrophages could determine their polarization. The LPS-treated macrophages from UAB126-fed diabetic mice showed an increased concentration of the anti-inflammatory IL-10 cytokine, which in general is known to be an M2 marker [37]. This result implies that during the course of diabetes, infiltrating monocytes may have become M2-type macrophages in the UAB126-treated diabetic retina as compared to the untreated. In line with the upregulated ECAR rate, we also observed reduced glucose consumption in UAB126-treated diabetic macrophages. While the mechanism by which UAB126 reduces serum glucose remains to be elucidated, the effect may be indirect through modulation of inflammatory signaling, which in turn could influence the insulin pathway [38]. However, the interpretation of the studies needs to be considered in the context of T1D vs. T2D. The dual activation of RXR and PPAR by RXR agonists such as UAB126 supports the feasibility of UAB126 RXR agonists in the treatment of T2D [38]. Specifically, insulin receptors may be desensitized in the retinas of T2D mice. Thus, UAB126 could help balance glucose uptake by retinal cells, which improves insulin-controlled rhodopsin signaling and restores diminished retinal function [39].

In support of this hypothesis, we found that the 6-week treatment of diabetic mice prevents the decline of retinal function. These data suggest that the vision deficit in treated diabetic mice is slowed down. Moreover, the increase in the p-AMPK level in treated diabetic retinas indicates a metabolic shift towards homeostasis. It has been demonstrated that diabetes-induced inflammation strongly correlates with a reduced AMPK pathway, the downregulation of which leads to diminished SIRT1 activation [40]. AMPK stimulation is known to prevent photoreceptor cell degeneration in T1D mice [41]. Therefore, an increase in AMPK activity may indicate a reduced inflammatory response and diminished photoreceptor functional loss. This data correlate with the observed reduction in the decline of A-wave scotopic amplitudes in UAB126-treated diabetic mice vs. untreated ones. Given that BGL is reduced, insulin is not changed, and AMPK activity is increased in UAB126-treated animal tissues, it is possible that RXR-based therapy provides effects like metformin in T2D patients [42].

The clinical importance of eye drops containing effective drugs such as UAB 126 to correct DR has marked translational significance as a way of delivering bioactive compounds on a daily basis. One of the reasons for choosing topical administration of UAB126 is that it likely does not affect blood glucose levels in a manner similar to other topically applied drugs used to treat diabetic retinopathy [28,29]. In this case, the topical route reinforces the idea that the reported effects are directly attributable to the drug itself rather than improvements in the diabetic environment. The topical application of UAB126 successfully delivered the drug to the retina and increased the expression of genes responsible for lipid metabolism. This suggests that UAB126 application could normalize local lipid synthesis. Thus, the reduced expression of *Elovl2* and *Elovl4*, elongases required for the biosynthesis of long-chain polyunsaturated fatty acids, occurs in diabetic retinas [43,44]. Their increase could help slow down retinal pathogenesis by producing complex lipids known to be diminished in diabetic retinas overall [45]. Future studies should overcome the limitations of the present research and identify the modified lipid profile in UAB126-treated diabetic retinas. Moreover, the screening and validation of new efficient rexinoids should be continued to move the field towards their clinical application to treat DR. In our study, both rexinoid 3 and rexinoid 6 not only significantly increased all three RXR isoforms in the retina exposed to high glucose but also dramatically enhanced PPARα and PPARγ, which are known to control inflammation in diabetic retinas [46].

One of the key issues in contemporary biology and the field studying NRs is whether RXR provides a separable response and whether an RXR homodimer can function as a biologically relevant transcription unit. In our study, an increase in RXR was accompanied by an increase in PPARs and LXR, non-permissive RXR binding partners. Therefore, it is important to understand how the pleiotropic effects of rexinoids are carried out. Future research identifying binding partners activated by UAB126 treatment and their requirements to form an active RXR/NR heterodimer to provide a therapeutic effect should be conducted. These studies could not only overcome the limitations of the current research, but also provide a link between rexinoids and cellular signaling activated by UAB126 treatment.

In conclusion, our studies support the therapeutic potential of rexinoids to treat DR. In sum, our data indicate that systemic RXR-based therapy could benefit diabetic patients not only by reducing the BGL and therefore controlling DR progression, but also by altering local retinal metabolism and the inflammatory response.

## Figures and Tables

**Figure 1 cells-12-02361-f001:**
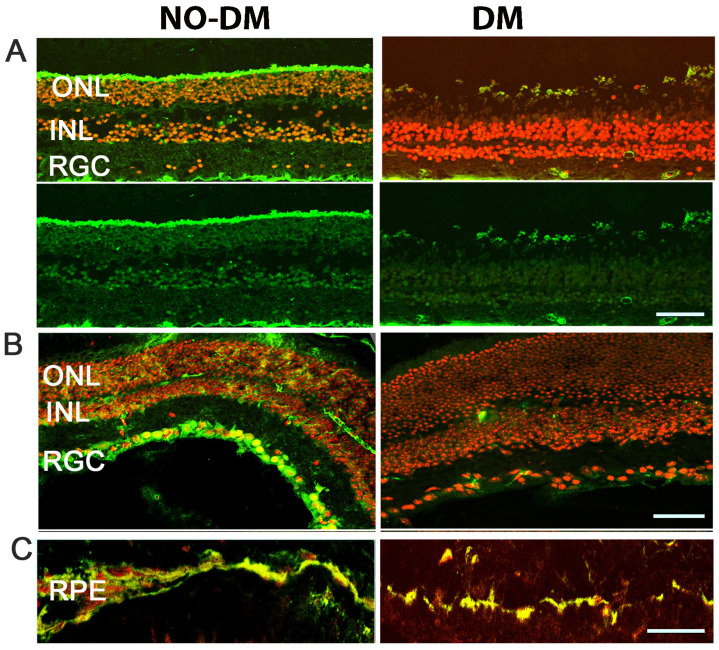
Expression of RXR in human and mouse diabetic retinas. (**A**) Immunostaining of human retinas comparing control (**left**) and diabetic (**right**) samples reveals reduced RXR expression (in green) in the outer nuclear layer (ONL), inner nuclear layer (INL), and retinal ganglion cell layer (RGL) of diabetic retinas. The nuclei are shown in red. (**B**,**C**) Immunostaining of db/db mouse retinas, a T2D mouse model, demonstrates decreased RXR expression in retinal ganglion cells (RGC), inner nuclear layer (INL), outer nuclear layer (ONL), and retinal pigment epithelial cells (RPE), consistent with the findings in human diabetic retinas. A scale bar is 50 μm.

**Figure 2 cells-12-02361-f002:**
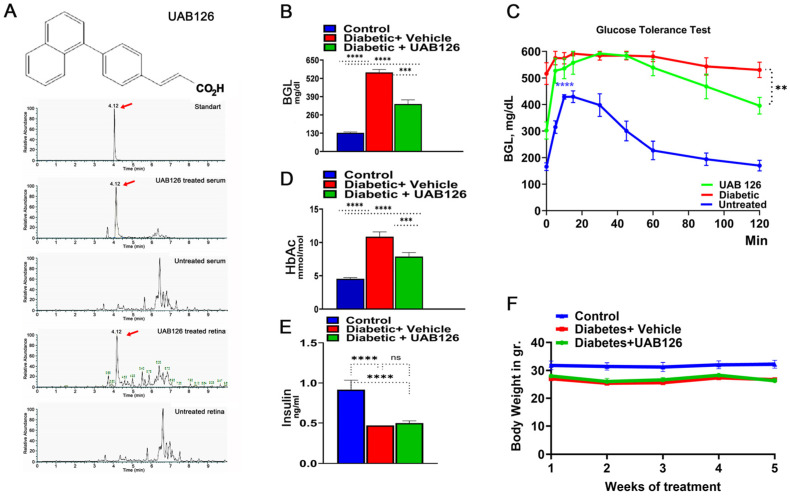
The oral administration of UAB126 in the diabetic retina. (**A**) LC-MS/MS analysis shows the detection of the RXR agonist, UAB126, in the serum and retinas of mice after systemic delivery. Representative LC-MS/MS peaks are shown in red arrows. (**B**) Diabetic mice treated with UAB126 for 6 weeks display a reduction in blood glucose levels (BGL) compared to diabetic controls (n = 3–6). The average of 6-week BGL measurements is shown. (**C**) The oral glucose tolerance test conducted in diabetic mice at the end of the UAB126 treatment; the UAB126 treatments reduced the blood glucose level in diabetic mice at 2 h after the oral glucose load, demonstrating the partial clearance of glucose from the body. Untreated diabetic mice, however, showed no difference in clearance. (**D**) The decrease in BGL correlates with HbA1C measurements, indicating the beneficial effects of UAB126 treatment on glycemic control at 6 weeks. (**E**) UAB126 treatment does not alter insulin levels, showing that the compound does not affect insulin secretion in diabetic mice. (**F**) The treatment of diabetic mice with UAB126 does not lead to significant body weight loss compared to untreated diabetic mice (n = 5). “ns”—not significant, ** *p* < 0.01, *** *p* < 0.001, and **** *p* < 0.0001.

**Figure 3 cells-12-02361-f003:**
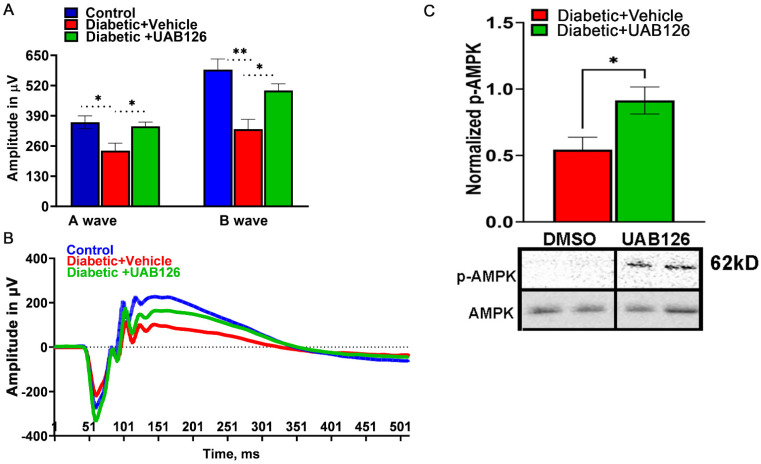
Treatment of diabetic mice with UAB126 prevents retinal function loss. (**A**) Electroretinogram (ERG) recordings from diabetic retinas show reduced A-wave amplitudes (photoreceptor responses) and B-wave amplitudes (bipolar and Muller cell responses) compared to healthy control retinas at 14 weeks after streptozotocin (STZ) injection. Treatment with UAB126 preserves the A-wave amplitude and partially protects the B-wave amplitude decline, indicating protection of retinal function. (**B**) Representative ERG waveforms from healthy control, diabetic + vehicle, and diabetic +UAB126-treated mice are shown. (**C**) The improvement in retinal function in UAB126-treated diabetic retinas is associated with an increase in AMPK activation (n = 5–6). * *p* < 0.05, ** *p* < 0.01.

**Figure 4 cells-12-02361-f004:**
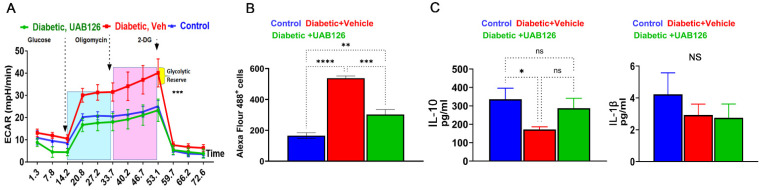
The UAB126 treatment normalizes metabolism in bone marrow-derived macrophages. (**A**) Glycolytic stress test measures the extracellular acidification rate (ECAR) at each step of glucose, oligomycin, and 2-DG addition (n = 3–4). No difference is observed between control and UAB126-treated macrophages. However, significant differences are observed between vehicle-treated diabetic and UAB126-treated diabetic or vehicle-treated and the control group at all steps of ECAR (*** *p* < 0.001). Results were analyzed by two-way ANOVA. (**B**) UAB126-treated macrophages show a lower rate of 2-DG (2-deoxyglucose) uptake compared to vehicle-treated macrophages, although not reaching the level of healthy untreated control macrophages. (**C**) UAB126-treated macrophages express increased levels of Il-10 mRNA, a biomarker of M2 macrophages, while Il-1β mRNA expression remains unchanged (n = 3–4). “ns”—not significant, * *p* < 0.05, ** *p* < 0.01, *** *p* < 0.001, and **** *p* < 0.0001.

**Figure 5 cells-12-02361-f005:**
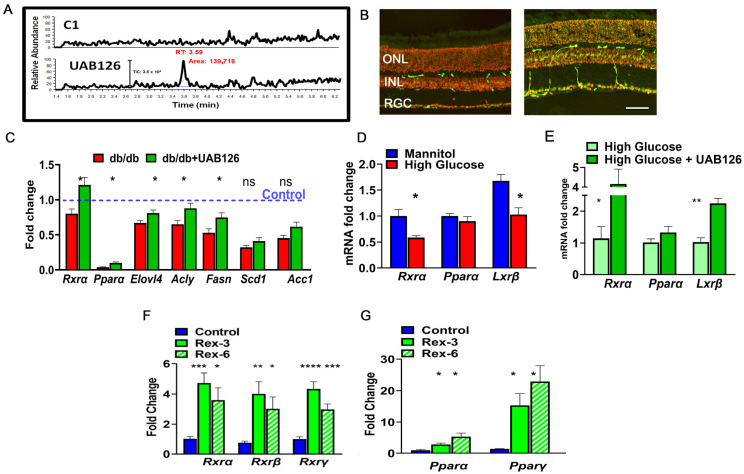
The eye drops containing UAB126 were designed for topical application and retinal delivery of rexinoids. (**A**) LC-MS/MS analysis confirms the accumulation of UAB126 in the retinas after topical application of eye drops. LC-MS/MS spectrograms show UAB126-treated retinas compared to untreated controls. (**B**) Immunostaining of control (**left**) and diabetic (**right**) retinas with anti-RXR antibody detects the RXR signal (in green) following topical UAB126 application. (**C**) Topical UAB126 application results in an increase in Rxrα, Pparα, and lipid-associated gene expression in db/db retinas, suggesting altered lipid metabolism in treated diabetic retinas (n = 3–5). (**D**) Retinal explants cultured in high glucose medium as compared to control (mannitol) exhibit decreased expression of Rxrα, Pparα, and Lxrβ mRNA. (**E**) Treatment with UAB126 in high glucose medium increases the expression of these genes (n = 3–4). (**F**) The application of other rexinoids (rexinoid 3 and rexinoid 6) to C57BL6 retinal explants exposed to high glucose results in a dramatic increase in all three RXR isoforms (n = 3–4). (**G**) In addition to RXR, UAB126 treatment induces an increase in *Pparα* and *Pparγ* mRNA expression, suggesting potential interactions and effects on other binding partners for RXR-based therapy (n = 4). “ns”—not significant, * *p* < 0.05, ** *p* < 0.01, *** *p* < 0.001, and **** *p* < 0.0001. A scale bar is 50 μm.

## Data Availability

The authors confirm that the data supporting the findings of this study are available from the corresponding author upon reasonable request.

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
