# Peer review of "Retinoid X Receptor Activation Prevents Diabetic Retinopathy in Murine Models"

_cells, 2023, doi:10.3390/cells12192361_

Round 1

Reviewer 1 Report

The ms is well designed and written. The topic is interesting for a disease at high medical needs,. The results convincing

Author Response

Dear Reviewer,

We appreciate your time and efforts in reviewing our manuscript.

Reviewer 2 Report

Dear author

Thank you for your research. After the decision regarding the preprint of the article and the addition of relevant text and references, it will be considered.

Author Response

Dear Reviewer,

We appreciate your time and efforts in reviewing our manuscript. We also would appreciate some clarification on the specific changes required in the methods section. Additionally, if possible, could the reviewer specify which references need to be replaced and suggest any additional references that should be included? This information will greatly assist us in making the necessary revisions. 

Reviewer 3 Report

In this paper the authors demonstrated the usefulness of RXR activation in preventing experimental DR. No major concerns in the design and the main results. However, the following issue should be commented in both the results and the discussion:

The systemic administration of UAB 126 significantly reduces the blood glucose levels, which could partly explain its the beneficial effects in preventing DR. By contrast, as previously published using other drugs with well-recognized hypoglycemic action (i.e PMID: 26384381, PMID: 28779212) the topical administration of UAB 126 is probably unable to reduce the blood glucose levels, but this information should be added to the results. This is important because the authors have seen beneficial effects also in this group treated with eye drops. Therefore, the confirmation of a lack of effect in lowering blood glucose levels by using the topical route will reinforce that the reported effects were directly related to the drug rather than to the diabetic milieu improvement. 

Author Response

Dear Reviewer,

We appreciate your time and efforts in reviewing our manuscript. We also appreciate your thoughtful comments and suggestions. In the revised manuscript, we included the requested information (highlighted in yellow).